# Exposomes to Exosomes: Exosomes as Tools to Study Epigenetic Adaptive Mechanisms in High-Altitude Humans

**DOI:** 10.3390/ijerph18168280

**Published:** 2021-08-05

**Authors:** Manju Padmasekar, Rajkumar Savai, Werner Seeger, Soni Savai Pullamsetti

**Affiliations:** 1Max-Planck Institute for Heart and Lung Research, Member of the German Center for Lung Research (DZL), Member of the Cardio-Pulmonary Institute (CPI), 61231 Bad Nauheim, Germany; Manjupadma-sekar.nandigama@mpi-bn.mpg.de (M.P.); Rajkumar.savai@mpi-bn.mpg.de (R.S.); werner.seeger@innere.med.uni-giessen.de (W.S.); 2Institute for Lung Health (ILH), Justus Liebig University, 35392 Giessen, Germany; 3Department of Internal Medicine, Justus-Liebig University Giessen, Member of the DZL, Member of CPI, 35392 Giessen, Germany; 4Frankfurt Cancer Institute (FCI), Goethe University, 60438 Frankfurt am Main, Germany

**Keywords:** high-altitude adaptation, high-altitude pulmonary hypertension, hypobaric hypoxia, epigenetics, exosomes

## Abstract

Humans on earth inhabit a wide range of environmental conditions and some environments are more challenging for human survival than others. However, many living beings, including humans, have developed adaptive mechanisms to live in such inhospitable, harsh environments. Among different difficult environments, high-altitude living is especially demanding because of diminished partial pressure of oxygen and resulting chronic hypobaric hypoxia. This results in poor blood oxygenation and reduces aerobic oxidative respiration in the mitochondria, leading to increased reactive oxygen species generation and activation of hypoxia-inducible gene expression. Genetic mechanisms in the adaptation to high altitude is well-studied, but there are only limited studies regarding the role of epigenetic mechanisms. The purpose of this review is to understand the epigenetic mechanisms behind high-altitude adaptive and maladaptive phenotypes. Hypobaric hypoxia is a form of cellular hypoxia, which is similar to the one suffered by critically-ill hypoxemia patients. Thus, understanding the adaptive epigenetic signals operating in in high-altitude adjusted indigenous populations may help in therapeutically modulating signaling pathways in hypoxemia patients by copying the most successful epigenotype. In addition, we have summarized the current information about exosomes in hypoxia research and prospects to use them as diagnostic tools to study the epigenome of high-altitude adapted healthy or maladapted individuals.

## 1. Introduction

Exposomes refers to the totality of environmental exposure individuals experience over their lifetime [1]. Our genome is influenced by exposomes for better or worse. Living organisms at high-altitudes are exposed to an entirely different set of exposomes when compared to lowlanders and this influences their epigenetic make-up. The understanding that organisms are capable of genetically ‘reprogramming’ themselves as well as ‘preprogramming’ future generations to optimize gene expression for a particular environmental signal may have tremendous impact on our perception of both acclimatization and adaptation to hypoxia [2]. High altitude is usually referred to elevation over 2500 m above sea level. Although the percentage of oxygen in air is constant at different altitudes (20.9%), as the vertical height above the earth’s surface increases, the atmospheric pressure (barometric pressure) decreases resulting in decreased partial pressure of inspired oxygen (pO_2_), leading to less oxygen molecules per breath. For example, at 4000 m altitude, one breath contains just 62% of the oxygen molecules present at sea level. Due to diminished pO_2_, the driving pressure for gas exchange in the lungs decreases, leading to poor oxygenation of blood as it passes through the pulmonary capillaries. This is more evident during strenuous activity when cardiac output increases. Hence, the blood spends less time at the gas exchanging surface, limiting the time for oxygen diffusion [3]. If an individual from lowland is acutely exposed to high-altitudes (>5500 m), they may lose consciousness, and at over 8000 m, it is an almost certain occurrence. However, when a person is gradually exposed to high-altitudes, they can acclimatize, adapt and survive. During this short-term adaptation phase, the body makes a wide variety of changes to tackle the decreased oxygen in blood [4].

Human survival depends on oxygen homeostasis, which is severely challenged during hypobaric hypoxia. Around 2% of people worldwide, more than 140 million people, live permanently at high altitudes of over 2500 m, such as the Ethiopian Highlands in Africa, the Himalaya Mountains in Asia, and the Andes Mountain Range in South America, where pO_2_ is low, UV radiation is high, and temperatures are low [5]. Such indigenous populations have developed adaptive features in their morphology, and also in physiological, biochemical, and molecular regulatory pathways to thrive in hypoxic, high-altitude environments. They show better tolerance to hypoxia and exceptional physical performance at altitude [6,7,8]. However, in some, this physiological adaptation fails and hypoxia triggers in them a maladaptive response that leads to various forms of acute and chronic high-altitude illness, such as high-altitude pulmonary edema or chronic mountain sickness [9]. Chronic mountain sickness or Monge’s disease was first described by Carlos Monge in the Andes in 1925 and is the result of failure to physiologically adapt to high-altitude exposure. Chronic mountain sickness occurs in natives or lifelong high-altitude residents, and is characterized by excessive erythrocytosis and severe hypoxemia. Oftentimes, it is associated with moderate or severe pulmonary hypertension (PH) that may evolve to right ventricular hypertrophy and dysfunction, which ultimately leads to congestive heart failure [10,11]. High-altitude pulmonary hypertension (HAPH) is characterized by a mean pulmonary artery pressure >30 mmHg and/or a systolic pulmonary artery pressure >50 mmHg in the absence of excessive erythrocytosis or other lung diseases and might be as frequent as 14% within the population [12]. In cattle, HAPH is known as brisket disease and can be inherited [13,14]. Developmental adaptation to high altitude that happens in natives is different from physiological acclimatization that occurs in sojourners. Native highlanders experience physiological changes to low oxygen for a lifetime, wherein it occurs over days to weeks in the latter. Notably, developmental adaptation is imprinted in the genome and is carried on to future generations, whereas acclimatization to high altitudes is short-lived and is reversible upon return to low altitudes [15].

The lungs play an important role in the acclimatization (short-term adaptation) of the human body to low oxygen at increased altitudes. During acute exposure of lungs to altitude, there is extravascular fluid accumulation in the pulmonary interstitium, and hence, a reduction in lung volume. In this scenario, hyperventilation, together with increased heart rate, helps to supply adequate oxygen to tissues. At rest, increased oxygen demand was met by firstly increasing the tidal volume, at least up to 3500 m [16,17]. Pulmonary circulation responds to hypobaric and normobaric hypoxia by increasing pulmonary arteriolar resistance, which maintains the ventilation–perfusion ratio during localized alveolar hypoxia [18]. Hypoxic pulmonary vasoconstriction occurs in small pulmonary arterioles and veins of a diameter of <900 µm, the veins accounting for ~20% of the total increase in pulmonary vascular resistance caused by hypoxia [19]. The magnitude of hypoxic pulmonary vasoconstriction differs among humans and this difference could be attributed to genetics and adaptive mechanisms. Chronic exposure to hypoxia as a result of high-altitude living can predispose certain individuals to HAPH due to adaptive mechanisms becoming exaggerated, leading to pulmonary vascular remodeling and development of PH, which places an increased pressure load on the right ventricle leading to right heart failure [20,21]. It must be emphasized that causes other than hypoxia may potentially form the basis of and/or contribute to HAPH, such as chronic heart and lung diseases, thrombotic or embolic disease and some gene mutations or even epigenetic mechanisms, which can actually exacerbate the burden on the already hypoxia exposed lungs. Unfortunately, at present, there are no clinically approved drugs for the therapy of HAPH, although the pathological burden is high [9].

Hypoxemia is not only present in high-altitude dwellers but also amongst critically ill patients, such as in PH associated with lung diseases or patients suffering from viral infections such as SARS-COV-2; yet, optimal management strategies remain uncertain. Mechanisms that lead to beneficial adaptation, as opposed to maladaptation, need to be dissected in order to develop individualized treatment strategies. Identifying genomic and epigenomic differences between lowlanders and those adapted for many centuries to the hypoxia of high-altitude may identify beneficial mechanisms of adaptation for subsequent evaluation in the clinical setting. Studying the physiology of members of indigenous populations who are well-adapted to high-altitude may reveal novel target pathways that are amenable to pharmacological manipulation in the critically ill, because trying to mimic the most effective human hypoxia-tolerant phenotype could provide new directions in identifying biomarkers and metabolic pathways which can be further exploited for patient benefit, and this knowledge can be renewed repeatedly as new technologies evolve.

Genomic variation plays an important role in living organisms’ adaptation to varied environments. Genome-wide association studies (GWAS) have identified several genes that underlie high-altitude adaptive phenotypes, many of which are central components of the transcription factor Hypoxia Inducible Factor (HIF), which plays a pivotal role during hypoxia. Candidate genes which are of positive selection among humans in high-altitude environments are *NOS2A* (nitric oxide synthase 2), *ADRA1b* (alpha-1B-adrenergic receptor), *EDN1* (endothelin 1), *PHD3* (HIF-prolyl hydroxylase 3), *VEGF* (vascular endothelial growth factor), *TNC* (tenascin C), *CDH1* (cadherin 1), *EDNRA* (endothelin receptor A), *PRKAA1* (protein kinase, AMP-activated, alpha 1 catalytic subunit), *ELF2* (E74-like factor 2), and *PIK3CA* (phosphoinositide-3-kinase, catalytic, alpha polypeptide), including the enzyme EGLN1 (Egl-9 Family Hypoxia Inducible Factor 1), which catalyzes the post-transcriptional formation of 4-hydroxyproline. EGLN1 downregulates HIF targets, including EPO (erythropoietin), which is involved in red blood cell production. Other candidate genes are *EPAS1* (endothelial PAS domain protein 1), *CYP2E1* (a cytochrome P450 enzyme), *EDNRA* (Endothelin Receptor Type A), *ANGPTL4* (angiopoietin-like 4), and *CAMK2D* (calcium/calmodulin-dependent protein kinase II delta) [22,23,24,25,26].

While genetic variations play an important role in adaptation to high-altitude and provides high-altitude dwellers with a survival advantage, not much is studied regarding the role of epigenetics in this context [27]. The study of any potentially stable and, ideally, heritable change in gene expression or cellular phenotype that occurs without changes in ‘Watson-Crick base-pairing of DNA’ is termed epigenetics [28]. The important points when defining epigenetics are heritability and reversibility [29]. Similar to DNA modifications, epigenetic imprints are also faithfully transmitted to offspring and is used as a means of communicating information to the daughter cell so that they remember what kind of cell it is apart from positional and signaling cues. For example, a skin and heart cell carry the same genetic information, yet differ in their gene expression pattern because of their unique epigenetic marks and, hence, they are able to faithfully copy the gene expression pattern of their parent cell. Epigenetic imprints persist through thousands of cell divisions for the lifetime of the organism, unless they are actively erased or lost through epimutation [30,31]. Epigenetic marks are established during early development shortly after fertilization and before preimplantation, DNA methylation is one such well-studied epigenetic modification which plays a prominent role in transcriptional regulation, silencing of repetitive DNA elements, and genomic imprinting. During early embryonic development, maternal and paternal genomes methylation pattern is erased and a fetal methylome is established [32]. Hence, the epigenome is more vulnerable to environmental insults during early development. Although accumulating studies show that epigenetic memory is transmitted across generations, the exact mechanism how it happens is still not clear, and it is an interesting area of research which is open for exciting discoveries [33,34]. In this review, we talk about the epigenetic mechanisms involved in the adaptation of living organisms to high altitude and discuss how exosomes can be excellent tools for understanding the epigenomics at work at high altitudes.

## 2. Epigenetics of Hypoxia

Hypoxic stimulus changes DNA methylation and post-translational histone modifications status of gene promoter or enhancer regions, leading to altered interaction of them with transcriptional machinery. On the other hand, an abundant of non-coding RNAs, such as microRNAs (miRNAs) and long non-coding RNAs (lncRNAs), either transcriptionally silence or degrade targeted messenger RNAs. These epigenetic modifications cause a change in gene expression without a change in DNA sequence, thus forming an integrated and highly complex regulatory network to create a gene expression pattern which forms the basis of hypoxic adaptation [35,36].

When the metabolic demand for oxygen exceeds the supply, hypoxia ensues. Hypoxia can arise under many pathological states, including inflammatory, fibrotic, ischemic and tumorigenic processes, and also due to environmental conditions, such as high-altitude dwelling. Earlier studies on high-altitude adaptations were focused on phenotypes such as hemoglobin concentration or physical work capacity. The subsequent discovery of HIF [37] has been a breakthrough in our understanding of adaption to high-altitudes. HIFs are the master regulator of hypoxic response and was first identified when investigating the molecular mechanisms of the induction of the hematopoietic growth hormone EPO during hypoxia [38]. HIF is a heterodimeric DNA-binding complex composed of two basic helix-loop-helix proteins, the constitutive HIF-1β and one of either hypoxia-inducible *α*-subunits, HIF-1α or HIF-2α (also known as EPAS1). HIF-β subunits are non-oxygen-responsive nuclear proteins, while HIF-α subunits are highly inducible by hypoxia. Under normoxic conditions, HIF-1α is degraded through hydroxylation, and hence, it is short lived [39]. However, during hypoxia, the α/β heterodimer binds to a core pentanucleotide sequence (RCGTG) in the hypoxia-response elements (HREs) of target genes, resulting in transcriptional regulation [37,40]. In that way, activated HIF-1α directly or indirectly controls the expression of more than 100 genes, and thus, mediates a wide range of physiological and cellular mechanisms necessary to adapt to reduced oxygen [41].

HIF and epigenetic events work together in coordinating hypoxic response pathways and also in the maintenance of the post-hypoxic phenotype. Epigenetics play four key roles during hypoxia: (i) epigenetic modifiers modulate HIF-1α mRNA or protein stability, such as hypermethylation of HIF regulators—von Hippel-Lindau (VHL) and PHD3—having a direct impact on hypoxic signaling, (ii) epigenetic modifiers, via interacting with HIF or modifying HRE binding sites, may also fine-tune HIF-dependent transcriptional programs; for example, HIF-1α associates directly with CREB-Binding-Protein/p300, and thus, augments HIF-1 transcriptional activity and participates in the co-activation of hypoxia inducible genes [42], (iii) the interplay between HIF and epigenome is bi-directional, as hypoxia/HIF leads to a significant global change in histone modifications and DNA methylation in the genome in response to hypoxic exposure, and (iv) the expression of different histone-modifying enzymes are also found to be direct HIF-1 target genes [43].

Notably, when cells were permanently maintained in hypoxia (1% O_2_), a global change in histone acetylation and DNA methylation was observed. Specifically, H3K9 acetylation and DNA methylation increased in the absence of HIF-1α along with a significant increase in the expression of DNA methyl transferase 3b (DNMT3b) [44]. Hypoxia induces a novel signature of histone modifications, including an increase in histone 3 lysine 4 trimethylation (H3K4me3) levels and a decrease in histone 3 lysine 27 trimethylation (H3K27me3) levels at the promoters of hypoxia-responsive genes and was also found to bind to genes which are marked with histone H3 lysine4 trimethylation (H3K4me3) [45,46]. Apart from global epigenetic changes, locus-specific epigenetic modulations of HIF regulators, such as hypermethylation of VHL and PHD3 promoters, are seen in certain tumor cells, which in turn amplifies HIF signaling [47,48]. Chronic hypoxia was shown to significantly induce methylation at the CpG site in the *VHL* promoter, decreased VHL expression, and increased EPAS1 and EPO expression, leading to excessive erythrocytosis in a rat model of chronic mountain sickness. Furthermore, the DNMT inhibitor 5-azacytidine reduced chronic hypoxia-induced erythroid proliferation in the bone marrow of rats with chronic mountain sickness by suppressing VHL methylation and DNMTs expression [49]. Chronic hypoxia was also found to upregulate DNMTs in ovine uterine arteries and thereby repressed large conductance Ca^2+^-activated K^+^ channel function [49].

On the other hand, VHL was shown to prevent HIF activation through the recruitment of histone deacetylase enzymes (HDACs) [49]. HDACs 1, 3, and 7 directly bind to HIF-1α, and thus, regulates its function during hypoxic exposure [50]. Sirtuin 1 (SIRT1) is a multifaceted NAD+-dependent protein deacetylase whose expression is increased during hypoxia [51]. SIRT1 modulates cellular responses to hypoxia by deacetylating HIF-1α and EPAS1, which leads to totally different outcomes: repression of HIF-1α activity and activation of EPAS1 signaling, respectively [52,53]. SIRT1-mediated deacetylation also suppresses the functions of tumor suppressor p53, which is stabilized and activated during hypoxic conditions. The p53KO mice was prone to pulmonary vascular remodeling and increased pulmonary arterial smooth muscle cell proliferation when exposed to chronic hypoxia leading to PH [54]. Hypoxia induced upregulation of HIF-1α is also found to activate Jumonji proteins (JMJDs), which possess histone demethylase properties [55]. It was also shown that high density methylation of the 5′-untransplated region (5′-UTR) of EPO controls its tissue specific expression and hypoxia activates Histone H3K9 demethylase JMJD1A leading to EPO expression [56,57]. Several histone lysine demethylases (KDMs) are modulated under hypoxia upon HIF-1α stabilization.

MicroRNAs are small RNA molecules, ∼22 nucleotides long, that can negatively control their target gene expression post-transcriptionally [58]. Recent evidence suggests that miRNAs are key elements in the response to hypoxia, regulating gene expression through post-transcriptional mechanisms. MicroRNAs are also important modulators of DNMT3a and DNMT3b expression under hypoxic conditions. A decreased expression of miR-29 correlates with increased DNMT3 expression in lung cancer cells [59], and it was shown that miR-29 levels fall significantly in ischemic hearts [60]. This might indicate that miR-29 controls DNMT3 expression under hypoxic conditions. miR-210 (hypoximir) is a well-known miRNA induced under hypoxic condition in several types of tissues and cells, and its targets control mitochondrial metabolism, angiogenesis, DNA damage response, apoptosis, and cell survival, and contributes to cellular adaptation to hypoxia [61,62]. These observations suggest the involvement of epigenetic processes in HIF-1-associated gene signaling in response to hypoxia, and may also control other unknown genomic processes. Thus, we can safely presume that exposure to high altitude may also trigger similar epigenetic mechanisms, leading to the adaptation of living beings to continuous hypobaric hypoxia. Accordingly, whole-genome DNA methylation data generated from heart tissues of Tibetan pigs grown in the highland and lowland, as well as Yorkshire pigs grown in the highland and lowland, using methylated DNA immunoprecipitation sequencing, showed several thousand differentially methylated regions (DMRs) containing many differentially methylated genes. Enrichment of DMRs associated with HIF-1 signaling pathway and pathways involved in hypoxia-related processes has been observed. Hence, they were considered to be important candidate genes for high-altitude adaptation in Tibetan pigs [63]. Apart from miRNAs, lncRNAs role in hypoxic regulation is increasingly evident in the field of cancer biology. lncRNAs do not code for proteins and are more than 200 nucleotides in length; they act in many different ways to modify gene expression. In cancer, they are found to act as enhancers of HIF-1α and EPAS1 expression and as well regulators of HIF stability and transcriptional activity [64].

## 3. Epigenetic Adaptation of Animals to High Altitude

Although the mechanism of high-altitude adaptation which may involve complex multi-gene expression and interaction is not fully understood, modern multi-omics approaches allow for the identification of key functional genes, and their regulatory mechanisms including epigenetic modifications underlying high-altitude adaptation (Figure 1). High-altitude adaptation is a classic example of selective pressure shaping the human genome [65]. Genetic variations alone cannot explain all the biological variations [66] that leads to adaptive or maladaptive processes that happen in high-altitude dwellers. DNA methylation patterns established prenatally as well as early in the postnatal period are associated with methylation patterns later in life, and thus, they may be involved in the organism’s phenotypic response to their environment. This phenomenon is known as the predictive-adaptive response, which helps the cells to be ‘prepared’ long after the stimulus which caused the epigenetic mark is gone, thus providing cells with a memory about its previous experiences [67]. This provides the cells with an ability to adapt to a new environment when genetic variation is limited. Hence, epigenetic changes can potentially act as mechanisms of phenotypic plasticity [68] through altered biological pathways that improve reproductive fitness and survival under conditions of chronic hypoxia, and studying them may provide a means of identifying candidate biochemical markers of successful adaptation to high altitude, and thereby, lead to viable pharmacological interventions that may benefit hypoxemic patients.

Animal models which are well-adapted to their high-altitude environments provide valuable information about the molecular mechanisms of hypoxic adaptation. Transcriptomic and proteomic analyses of chorioallantoic membrane tissues were performed in Tibetan and Chahua chicken embryos under hypoxic incubation using RNA-sequencing (RNA-Seq) and Isobaric tags for relative and absolute quantitation (iTRAQ). The Tibetan chicken is a unique breed that has adapted to the high-altitude hypoxic conditions of the Tibetan plateau and the Chahua chicken is domesticated from the red jungle fowl, which is a low-altitude typical local chicken breed in Yunnan Province. The obtained results showed 160 differentially expressed genes and 387 differentially expressed proteins that were mainly enriched in angiogenesis, vasculature development, blood vessel morphogenesis, blood circulation, renin-angiotensin system, and HIF-1 and VEGF signaling pathways. Twenty-six genes involved in angiogenesis and blood circulation, two genes involved in ion transport, and six genes that regulated energy metabolism were identified as candidate functional genes in regulating hypoxic adaptation of chicken embryos [69].

Histone acetyltransferase p300 (*EP300*) is a candidate gene showing relatively deep allelic divergence between Tibetans and lowlanders. EP300 enhances the expression of HIF-1α by modifying the chromatin, which in turn leads to increase in nitric oxide (NO) generation, as the NO gene contains HRE. Increased NO levels would allow for better vasodilation and, therefore, better blood flow and high NO levels are an adaptive feature of Tibetans for high-altitude living. It was also shown that *EP300* variants which are present in Tibetans lead to reduced blood NO, which is contrary to our understanding. NO also helps create a blunted response to hypoxia by inhibiting the oxygen consumption of mitochondria, and consequently, this provides more oxygen for PHDs to reduce the levels of HIF proteins caused by hypoxia; this, in turn, reduces blood NO. Thus, the adaptive alleles of *EP300* were associated with a decreased level of blood NO, which might serve as a protection measure for Tibetans from overproduction of NO [70].

Epigenetic variation, including histone modifications, RNA-based mechanisms, and DNA methylation, plays a crucial role in regulating gene expression in response to varied environmental cues. It was shown that incremental exposure to hypoxia can affect the epigenome [27]. Changes to the epigenome, in turn, can help with high-altitude acclimatization and adaptation. Similarly, in high-altitude dwelling humans, exposure to hypoxia could result in epigenetic alterations of their genome. Out of many epigenetic modifications, DNA methylation is the most studied in the context of hypoxia. The methylation pattern of the genome changes when living organisms are exposed to hypoxia. High-altitude hypoxia was shown to increase global levels of DNA methylation in Peruvian Quechua measured at the Long Interspersed Nuclear Element-1 (LINE-1). LINE-1 is a repetitive DNA element that constitutes about 21% of the human genome. Its high genomic frequency and its association with environmental exposures and disease-state make LINE-1 a suitable genetic marker to study the effects of high-altitude hypoxia on the epigenome. DNA methylation changes are also used to estimate the biological age of an organism and is referred to as DNAmAge. There are specific cytosine-phosphate-guanine sites (CpGs), which are either hypermethylated or hypomathylated with age and can be used to estimate the biological age of a living organism using some mathematical alogarithms [71], and these predictors are called epigenetic clocks or DNAmAge predictors, with the unit in years. In vitro experiments with human fibroblasts showed that the progression of DNAmAge was slower under hypoxia (1% oxygen) compared to normoxia (21% oxygen) [72]. On the contrary, in humans, high-altitude exposure was found to accelerate aging as predicted using epigenetic clocks [73].

Andeans, who have been high-altitude residents for millennia, showed decreased gene-specific *EPAS1* methylation. *EPAS1* is a hypoxia-responsive transcription factor that has been shown to be under selection in high-altitude Tibetans and Andeans. Its expression is higher in hypoxic conditions, and it is found to differentially methylated among Andean Quechua, suggesting that epigenetic modifications may play a role in high-altitude adaptation [74]. *EPAS1* seems to make Tibetans less likely to overproduce red blood cells at extreme altitudes and, hence, probably helps them to avoid altitude sickness and deliver oxygen more effectively to developing fetuses, thus contributing to their altitude aptitude [24]. Andeans are one of the best studied high-altitude adapted populations. They have inhabited the Andean Altiplano (average elevation of 3650 m) for up to 11,000 years, and hence, they display unique physiological adaptations to tackle hypobaric hypoxia, such as increased chest circumference which can accommodate greater lung volumes, elevated hemoglobin concentration, elevated arterial oxygen saturation, and a blunted (low) hypoxic ventilatory response [75]. Nevertheless, not all highlanders have adapted to high-altitude selective pressure in a similar way. Different indigenous populations who occupy high-altitudes exhibit differences in resting ventilation, hypoxic ventilatory response, oxygen saturation, and hemoglobin concentration, with an ultimate outcome of effective oxygen usage and survival at high-altitudes [76]. This differential adaptation to a similar stimulus could be explained partly by distinct epigenetic responses. For example, differences in the DNA methylation pattern among major ethnic groups has been observed [77]. Genome-wide DNA methylation studies in high-altitude adaptation is a great technique which will help in determining the individual genes and pathways that may undergo changes in DNA methylation in association with adaptation to high-altitude [73,78]. Epigenome-wide DNA methylation association study based on whole blood from 113 Peruvian Quechua with differential lifetime exposures to high-altitude (>2500 m) showed two significant differentially methylated positions (DMPs) and 62 differentially methylated regions (DMRs) associated with high-altitude developmental and lifelong exposure statuses. DMPs and DMRs were found in genes associated with hypoxia-inducible factor pathway, red blood cell production, blood pressure, anaerobic glycolysis, and others [73]. One of the two DMPs was located in the coiled-coil domain containing 64 (*CCDC64*) gene, which is involved in neuronal development by regulating the transport of secretory vesicles. *CCDC64* was identified as a candidate for genetic adaptation to high altitude among Tibetans and has been proposed to be under convergent evolution to high altitude in Asia and South America [23,79]. The other DMP was located in the gene *PAX7* involved in skeletal development and probably plays a role in the adaptation to physically demanding jobs at high altitudes. These findings suggest that being born at high altitude leaves a persistent mark on the epigenome.

Small non-coding mRNAs, such as miRNAs, play important roles in cold acclimation [80], hypoxia stress, and ultraviolet radiation [81] stress. To reveal the potential roles of miRNAs in high-altitude adaptation, high-quality small RNA sequencing data for six tissues (heart, liver, spleen, lung, kidney, muscle) from high- and low-altitude populations of three species, i.e., chicken, sheep, and pig, were generated and several orthologues and some novel miRNAs were found [82].

Yan et al. showed that the expression profiles of plasma miRNAs from newly arrived migrant Han Chinese (Tibet Han) residing at 3560 m were expressively different from those of the Nanjing Han lowlanders, with 172 differently expressed miRNAs. Although most of the differently expressed miRNAs did not show any statistically significant difference between native Tibetan and Tibet Han individuals, the plasma absolute concentrations of miR-210-3p were also significantly higher in the Tibet Han and Tibetan groups than in the Nanjing Han group. Computational analysis revealed that some HIF pathway genes, such as the *EGLN1* gene, were potential target genes of some highlighted miRNAs, and the upregulated miRNAs showed strong positive correlations with hematological variables, suggesting that living at high altitude has a remarkable influence on miRNA expression [83]. Study showed that plasma miR-210-3p might partly originate from peripheral blood cells and it was speculated that miR-210-3p might mainly be secreted by the above-mentioned cells via cell-derived microvesicles or exosomes, or as microvesicles-free miRNAs into the circulation in response to hypoxic conditions, resulting in the increase of plasma miR-210-3p level.

Sherpas, an ethnic group from the mountain regions of Nepal, are well-known for their exceptional physical strength at high-altitudes. They have adapted to high altitude so well that little acute or chronic mountain sickness has been documented in them. Adaptive physiological characteristics of Sherpas are determined by genetic selection due to environmental pressure combined with complex interaction of multiple genes and epigenetic processes. It has been already reported that the skeletal muscle cross-sectional area, mitochondrial density, and mitochondrial coupling efficiency are different between high- and lowlanders [84]. The skeletal muscles of Sherpas show lower force reduction during the fatigue-inducing protocol due to the lower accumulation of reactive oxygen species (ROS). There have been several studies over the past few years that identified redox-sensitive miRNAs and miRNA that targets redox regulating molecules such as Nuclear factor erythroid 2-related factor 2 (Nrf2), SIRT1, and Nuclear factor-κB (NF-κB) [85,86,87]. In addition, several studies have revealed that miRNAs via regulating several redox signaling pathways could modulate ROS production [88], indicating possible involvement of epigenomics in the superior skeletal muscle function in Sherpas. Six upregulated miRNAs and 20 downregulated miRNAs were identified by analyzing the miRNA expression profiles of rats exposed to high-altitude hypoxia when compared to rats housed at normal conditions. Six upregulated miRNAs were found to control Forkhead box O (FoxO), cGMP- protein kinase G (PKG), and p53 pathways and might play a critical role during hypobaric hypoxia exposure [89].

The role of lncRNAs in high-altitude adaptation has not been studied much, and only a few studies currently exist. One such study screened for lncRNAs from the gluteus transcriptomes of yak and their transcriptional levels were compared with those in Sanjiang cattle, Holstein cattle, and Tibetan cattle. Muscle tissues require a large amount of oxygen to function optimally. Under high-altitude condition, exercise capacity drastically decreased in non-native animals owing to low oxygen conditions [90]. Compared with cattle, yak’s muscle shows higher activities of lactate dehydrogenase (LDH), malate dehydrogenase (MDH), and β-hydroxyacyl-CoA dehydrogenase (HOAD), displaying a higher anaerobic potential in carbohydrate metabolism and a higher oxidative capacity, which indicates that yak might have developed special metabolism mechanisms in muscle tissues to adapt to high-altitude conditions. The study showed that compared with yak, 193, 361, and 433 lncRNAs were significantly differentially expressed in Tibetan cattle, Sanjiang cattle, and Holstein cattle, respectively. Transcriptional levels of myosin and EF-hand domain were increased in yak and are highly correlated with the lncRNAs MSTRG.25261.1 and MSTRG.2086.1. Over-representation of these genes should increase the binding ability to Ca^2+^ and, subsequently, enhance muscle contraction even at a low level of Ca^2+^. The transcription of tropomyosin might be regulated by lncRNAs MSTRG.1770.1 and MSTRG.24686 and transcription of titin by lncRNA MSTRG.11409.1. Tropomyosin and titin levels were downregulated in yak compared with the three cattle strains, which might facilitate the muscle contraction, since tropomyosin and titin plays an opposite role during the process of muscle contraction [91]. Whole-transcriptome analysis of yak and cattle heart tissues predicted several miRNAs/lncRNAs that are significantly enriched in high-altitude adaptation targeting T cell receptor signaling, VEGF signaling, and cAMP signaling [92] In Tibetan chickens, several key candidate Competing endogenous RNAs (ceRNAs) (DE lncRNAs-DE miRNAs-DE genes), which may play high-priority roles in the hypoxic adaptation of Tibetan chickens by regulating angiogenesis and energy metabolism, were identified [93]. A study which aimed to identify DE genes and novel lncRNAs of yaks for plateau adaptation and their underlying co-expression and regulatory network found a valuable sub-network comprising eight hub genes, one known lncRNA and five novel lncRNAs in the major module. These hub genes are associated with blood pressure regulation, generation of reactive oxygen species, and metabolism [94]. Not much is known about histone modifications in the context of high-altitude adaptation. Hopefully, future studies will explore this area of epigenetics in detail.

With the invent of high-throughput technologies, DNA methylation, acetylation, histone modifications/chromatin remodeling, and post translational RNA regulations are increasingly being studied and understood as arbitrators of crosstalk between genes and environment, thus helping to understand high-altitude adaptation and paving the way to epigenetic-based therapeutics for hypoxia-induced maladaptive phenotypes.

## 4. Epigenetics of High-Altitude-Induced (Hypobaric Hypoxia-Induced) Pulmonary Hypertension

PH is a complex, chronic, progressive, severely debilitating, incurable, and life-threatening disease of the vascular and non-vascular components of lung leading to hemodynamic alteration of pulmonary circulation. As a result, PH patients suffer from increased pulmonary arterial pressure due to increased vascular resistance, which eventually culminates in right ventricular dysfunction and failure due to pressure overload. PH occurs due to a variety of genetic and pathogenic causes and also due to the environmental conditions to which one is exposed [95] and is characterized by pulmonary vasoconstriction and abnormal (“pseudo-malignant”) inward remodeling processes that may affect all vessel layers (intima, media, and adventitia) [96,97].

Hypobaric hypoxia at high altitudes can contribute to chronic hypoxic stimuli for living beings present there and can lead to permanent pulmonary vascular remodeling and increased pulmonary vascular resistance, which manifests as a subgroup of PH known as HAPH [98]. HAPH occurs in populations who occupy mountainous regions where the pO_2_ is lower, leading to hypobaric hypoxia and hypoxemia. Not everyone living in high-altitude regions are prone to PH. For example, the prevalence of HAPH is 14–20% in the Kyrgyz population, the remaining being resistant or adapted [99]. HAPH patients show similar pathological remodeling of pulmonary vessels to their low-altitude counterparts, and exhibit enhanced pulmonary arterial pressure, which ultimately puts increased workload on the right ventricle leading to hypertrophy and heart failure. This form of PH is completely reversible when the person suffering with HAPH moves to low altitude, thus getting away from the hypoxic stimuli that are the cause for PH [100]. However, there also exists a rare form of irreversible, “out-of-proportion” severe PH, resulting in right heart failure and death, which represents more of a pathologic adaptation to hypoxia [101]. This implies that when the pathological stimuli are resolved, most HAPH is reversible. Studying epigenetic signatures of healthy highlanders and humans who developed HAPH will help in identifying dysregulated pathways and can pave the way for curative therapies both in HAPH and PH lowland patients.

Cells isolated from the vessel walls of PH lungs are found to be constitutively activated or “imprinted” towards a hyperproliferative, apoptosis-resistant, pro-inflammatory, and pro-fibrotic phenotype, even after being transferred outside the vascular microenvironment, demonstrating the acquisition of stable, functional phenotypic changes in mesenchymal cells, such as the fibroblast, and these changes probably require epigenetic processes which occur in response to altered histone acetylation, DNA methylation, and/or changes in micro-RNA expression profiles [102]. Although not much is known about the epigenetic modulations that happen in human models of HAPH, there are some studies which point to the epigenetic regulation that happen during HAPH both in humans and animal models.

Neonatal hypoxia was shown to increase global DNA methylation levels in lungs and specific cytosine methylation levels around the pulmonary Insulin-like growth factor 1 (IGF-1) promoter region in mice. HDAC inhibition with apicidin reduced chronic hypoxia-induced activation of IGF-1/Protein kinase B (AKT) signaling in lungs and attenuated right ventricular hypertrophy and pulmonary vascular remodeling [103].

The role of miRNA in PH has been identified in many studies [104,105], but the role of miRNA in HAPH is not well-known. A study showed that progressive hypobaric hypoxia was found to significantly affect levels of circulating miR-17, -21, and -190 in humans. Independently from the extent of hypoxemia, miR-17 and -190 significantly correlate with increased systemic pulmonary artery pressure [106]. Accordingly, plasma samples from cattle that developed PH at high-altitude showed higher miR-22-3 (miRNA that controls muscle function) in comparison to miR-451a (which controls erythrocyte function) when compared to cattle tolerant to high altitude, indicating a role of these miRNAs in HAPH [107]. Analysis of epigenetic alterations in the pulmonary vasculature of fetal lambs exposed to high-altitude long-term hypoxia (LTH) or to sea level atmosphere was performed. Intrapulmonary arteries were isolated, and fetal pulmonary artery smooth muscle cells (PASMCs) were cultured from both control and LTH fetuses. Compared with controls, in LTH fetuses, pulmonary arteries measurements of histone acetylation and global DNA methylation demonstrated reduced levels of global histone 4 acetylation and DNA methylation, accompanied by the loss of the cyclin-dependent kinase inhibitor p21. Treatment of LTH fetal PASMCs with histone deacetylase (HDAC) inhibitor trichostatin A decreased their proliferation rate, in part because of altered expression of p21 at both RNA and protein level. In the PASMCs of LTH fetuses, HDAC inhibition also decreased platelet-derived growth factor–BB (PDGF-BB)-induced cell migration and extracellular signal-regulated kinases (ERK1/2) activation and modulated global DNA methylation, substantiating the importance of epigenetic alterations in high-altitude-induced PH [108]. Similarly, Stenmark et al. showed a significant decrease of miR-124 expression in fibroblasts isolated from calves and humans with severe PH. Overexpression of miR-124 significantly downregulated the proliferation, migration, and monocyte chemotactic protein-1 (MCP-1) expression of hypertensive fibroblasts. Furthermore, the alternative splicing factor, polypyrimidine tract–binding protein 1 (PTBP1), was shown to be a direct target of miR-124 and was found to be upregulated both in vivo and in vitro in bovine and human pulmonary hypertensive fibroblasts. PTBP1, in turn, regulated the expression of Notch1/phosphatase and tensin homolog/FOXO3/p21Cip1 and p27Kip1 signaling. It was also shown that miR-124 expression suppressed HDAC and that treatment of hypertensive fibroblasts with HDAC inhibitors increased miR-124 expression and decreased proliferation and MCP-1 production [109].

Global DNA methylation was increased in PH rat models after hypobaric hypoxia exposure. DNMT3B was upregulated in both PH patients and rodent models. Furthermore, *Dnmt3b*^−/−^ rats exhibited more severe pulmonary vascular remodeling and inhibition of DNMT3B promoted proliferation/migration of PASMCs in response to PDGF-BB. In contrast, overexpressing DNMT3B in PASMCs was found to attenuate PDGF-BB-induced proliferation/migration and ameliorated hypoxia-mediated PH and right ventricular hypertrophy in mice. DNMT3B was also shown to transcriptionally regulate inflammatory pathways [110].

PH is also a hallmark of High-Altitude Pulmonary Edema (HAPE). Aberrant methylation in the apelin system could predict the risk of HAPE. HIF stimulates the secretion of apelin, a potent vasodilator during hypobaric hypoxia. HAPE patients show decreased apelin and nitrite levels as opposed to humans who are well-adapted to hypoxia. The increased methylation of the apelin CpG island, and hence, a lower secretion of apelin and, consequently, lower NO levels, could contribute to the pathology of HAPE due to impaired vasodilation [111].

The above-mentioned studies point to the probable role of epigenetic alterations in the genome of high-altitude living beings and in the pathology of HAPH. Contrasting to genetic mutations, epigenetic changes are pharmacologically reversible, making them an attractive target as therapeutic strategies for HAPH.

## 5. Exosomes and Hypoxia

Exosomes are the smallest type of extracellular vesicles (30–100 nm) released by almost all cells of the human body both in health as well as in diseased conditions [112]. During its initial discovery, they were thought to carry unwanted waste materials. However, later studies show that they act as messengers communicating between cells of different tissues by influencing the physiological and metabolic pathways of recipient cells [113]. Exosomes carry varied and useful cargo load, which includes cell membrane proteins, enzymes, growth factors, cytokines, DNA, mitochondrial DNA (mtDNA), mRNAs, lncRNAs, miRNAs, small nucleolar RNAs (snoRNAs), and lipids. They also carry signaling molecules and help in tissue remodeling [114]. The field of exosomes is a rapidly growing area in basic research, biomarker discovery, as well as in regenerative therapy, and even in nano biotechnology, as nanocarriers of drugs because of their viral-like transfection efficiency and inherent biological functions. Exosomes are enriched with molecules such as CD63, CD9, and CD81, Alix, and TSG101, which serve as markers for exosomes; however, the distribution of these surface molecules varies greatly depending on the cell type [112]. The exosomes are also a source of several potential biomarkers that are well correlated with the disease’s severity [115].

The lungs are a complex organ with a wide variety of cell types. Hence, exosomes are thought to play an important role in lungs during cell–cell communication. The lungs are also the organ with the highest vascular density in human body. It is very much understandable that the lungs contribute to the majority of exosomes that are circulating in the blood. Exosomes are released by a wide range of cell types present within the lungs, including endothelial cells, stem cells, epithelial cells, alveolar macrophage, and tumor cells [116]. During disease conditions, this exosomal load can be higher or lower depending on the stimuli, and the exosomal cargo may also vary widely depending on the infecting organism, immune system players, and the inflammatory microenvironment [117].

Exosomes contain a variety of RNA species, among which miRNAs are the most abundant and surely most intensively studied, while lncRNAs and circular RNAs (circRNAs) are also now becoming research hotspots [118,119,120]. Some qualitative and quantitative assays have revealed the asymmetric distribution of RNAs between cells and cell-derived exosomes. This phenomenon has boosted many interesting hypotheses, suggesting that RNA molecules are not randomly packaged in exosomes but with a set of sorting systems involved [121]. Gene ontology (GO) analysis has indicated that many mRNAs and proteins contained in extracellular vesicles are involved in epigenetic regulation, and the exosome-mediated transfer of miRNAs is considered to be an important mechanism of genetic exchange between cells [122]. Thus, extracellular vesicles regulate epigenetic processes, including DNA methylation, histone modification, and miRNA or lncRNA regulation, and the resultant epigenetic modifications are responsible for changes in the expression of tumor promoting genes and tumor suppressing genes. Recent research indicated that the detection of epigenetic biomarkers, such as miRNAs, in extracellular vesicles could be exploited for diagnosis of cancer or assessment of cancer prognosis [123]. The exosomal RNA cargo is unique, protected from degradation from nucleases, proteases, and oxidative stress, so that they can be delivered to recipient cells in an efficient manner, and they are also a concentrated source of information, which is used to alter the function of neighboring and distant cells, and they also carry a wealth of information about transcriptomic and epitranscriptomic changes that occur during disease conditions [124].

Hypoxia triggers the release of exosomes and influences the secretion, composition, and function of exosomes. Hypoxia also alters miRNAs, proteins, lipids, and metabolites loaded in exosomes. Exosomes are not yet extensively studied in the context of high-altitude dwellers’ adaptation and maladaptation to hypobaric hypoxia. However, these tiny yet powerful information carriers have evolved as major players in tumor biology, where hypoxia plays a major role, too. It was shown that when three different breast cancer cell lines were exposed to moderate (1% O_2_) and severe (0.1% O_2_) hypoxia, the number of released exosomes significantly increased [125]. Hypoxic cells express numerous plasma membrane receptors, such as glucose transporter (GLUT-1), epidermal growth factor receptor (EGFR), transfer receptors, P-glycoprotein (P-gp), and multidrug resistance protein 1 (MRP1), and this altered receptor expression pattern could act as a trigger for receptor activation and internalization or result in receptor clustering, leading to endocytosis and promoting exosome release [126]. Hypoxic stress not only results in increased exosome secretion, it can also lead to qualitative differences in exosomes, leading to differential bioactive cargo and function. Hypoxic exosomes was found to be enriched with proteins such as protein-lysine 6-oxidase (LOX), thrombospondin-1 (TSP1), and VEGF, and a disintegrin and metalloproteinase with thrombospondin motifs 1 (ADAMTS1), which are well-studied contributors to tumor progression, metastasis, and angiogenesis [127]. Hypoxic exosomes were also enriched with different non-coding RNAs [128,129]. In cancer cells, miR-210 is highly prevalent in hypoxic exosomes, which is expressed in an HIF-1α-dependent manner and is involved in navigating metabolic changes, angiogenesis, cell proliferation, migration, invasion, apoptosis, and stemness [130].

A search in PubMed with the search term “Exosomes and epigenetics” generated 439 results for the time period between 2005–2021, and out of those, 239 publications have come out only in the last two years (2019–2021). This indicates that the techniques for isolating and purifying exosomes have become more refined and doable. This shows that, combined with the improvement of high-throughput sequencing (HTS) technologies, the last couple of years have seen a robust increase in the field of exosomal epigenetics research. However, the search term “exosomes and high-altitude adaptation” came up with only one publication. This shows that there exists an enormous possibility to study exosomes in the context of high-altitude adaptation, and since exosomes are enriched with epigenetic information in the form of mRNAs, miRNAs, circRNAs, and lncRNAs, they will prove to be excellent tools to understand epigenomics of high-altitude adaptation and maladaptation. Studying exosomes has enormous potential for biomarker discovery and also in discovering beneficial pathways that can help with hypoxic adaptation.

## 6. Exosome for High-Altitude Epigenetic Research

Exosomes are secreted by most cells of humans and are found abundantly in body fluids, such as saliva, plasma, serum, cerebrospinal fluid, and urine [131,132,133,134,135]. Hence, exosomes can be isolated easily from plethora of body fluids—thus, they are easily sampleable. In cancer biology, liquid biopsies have revolutionized the field of biomarker discovery, and exosomes play a pivotal role [123,136,137]. Exosomes from liquid biopsies are extremely stable and their phospholipid bilayer protects their bioactive cargo from degradation [138,139]. DNA methylation appears to be cell-type specific, which poses challenge in human studies done using whole blood because sampling is usually limited to the analysis of peripheral blood cells, which may or may not reflect systemic adaptations. Exosomes can overcome this challenge as their isolation and analytic techniques has gotten more and more refined. Exosomes from serum or plasma also reflect the exosomal content from many different tissues and cells of body. However, advancement in exosomal biology allows us to identify and separate exosomes from a particular tissue by its membrane protein content [140], and hence, studying their genomic content can be more specific and fruitful in terms of understanding a specific organ’s epigenetic profile during health and disease rather than from whole blood.

Exosomes carry unique sets of mRNAs, rRNAs, miRNAs, lncRNAs, and other small non-coding RNAs (ncRNAs, e.g., piRNAs, snRNAs, snoRNAs, scaRNAs, and Y-RNAs) [141,142]. Some miRNAs that are abundant in parent cells are also most commonly found in exosomes, hence making them excellent tools for biomarker discovery. However, this is not always the case [141,143]. The majority of microRNAs detectable in serum and saliva were found to be concentrated in exosomes [144]. It has been reported that the absolute amounts of miRNAs are estimated to range 1 × 10^2^–1 × 10^5^ copies per cell [145], while a single exosome can carry up to approximately 500 copies of miRNAs [145]. With the current advances in next-generation deep sequencing (NGS), the entire spectrum of known and novel miRNA can be profiled with minimal RNA input and hence analysis of exosomal RNA has become much feasible [146]. The most important limitation in working with exosomes is their nanosize, and hence, the laborious and time-consuming ultracentrifugation steps that are involved in their isolation. However, recent advancements in the exosomal isolation kits that are available in market has decreased the burden of extensive ultracentrifugal steps without having to compromise exosomal quality [147]. The coming years might prove to be comparatively trouble-free and cost-effective as far as exosome isolation is concerned. International Society for Extracellular Vesicles provides guidelines to be followed for the publication of EV (extracellular vesicle) studies [148].

All this makes exosomes a good tool for epigenetic research, and it would be very helpful to discover epigenetic adaptive and maladaptive pathways in indigenous populations occupying high-altitude environments or in sojourners (Figure 2).

## 7. Conclusions

Although epigenetics of PH has been extensively probed and studied, the epigenetic mechanisms underlying high-altitude adaptation and HAPH have been less explored in human models. Studying hypobaric hypoxia-induced epigenetic changes in living organisms will provide us with clues about which signaling pathways go awry in hypoxemia patients. Since exosomes are available in almost all different biological fluids of an organism, isolating and studying their bioactive cargo in a non-invasive or minimally invasive manner is possible. Attempts to study exosomal bioactive cargo of high-altitude well-adapted and maladapted humans, or other organisms, may lead to valuable biomarker discoveries, as well as providing us with knowledge about novel signaling pathways and the possibilities of manipulating them with epigenetic drugs, which can lead to many exciting discoveries for treating diseases such as PH and HAPH and many more ailments that afflict the lungs.

## Figures and Tables

**Figure 1 ijerph-18-08280-f001:**
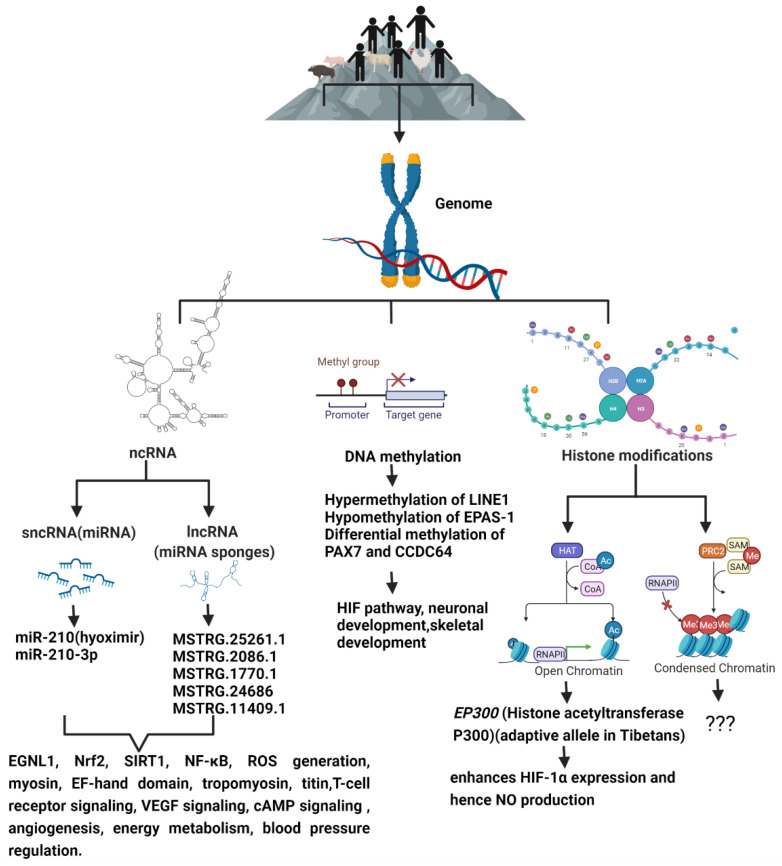
The schematic diagram shows the epigenetic modifications predominant in high-altitude living organisms when compared to their lowland counterparts, and the genes and signaling pathways that are altered as a result of high-altitude living.

**Figure 2 ijerph-18-08280-f002:**
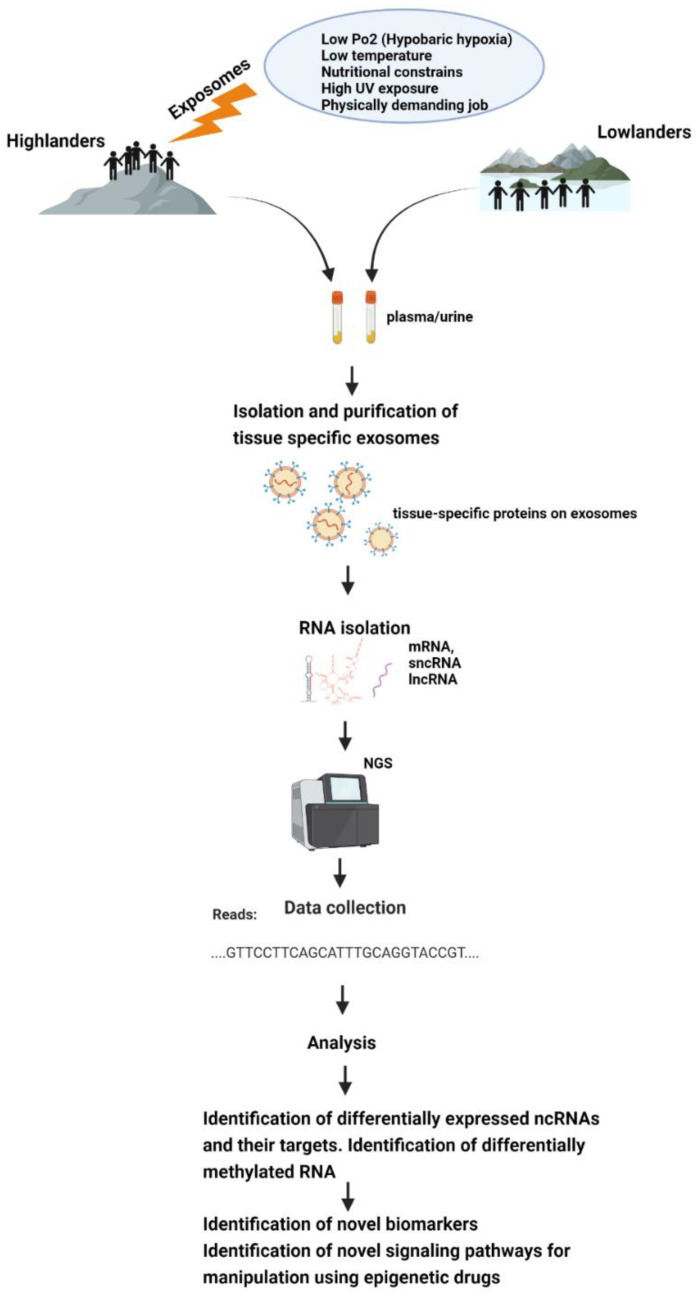
Schematic representation of how exposomes affect the epigenome of high-altitude dwelling living organisms—which could be studied using exosomes.

## Data Availability

Not applicable.

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
