# Peer review of "Exposomes to Exosomes: Exosomes as Tools to Study Epigenetic Adaptive Mechanisms in High-Altitude Humans"

_ijerph, 2021, doi:10.3390/ijerph18168280_

Round 1

Reviewer 1 Report

In their review "Exposomes to Exosomes: Exosomes as Tools to Study Epigenetic Adaptive Mechanisms in High Altitude Humans" the authors summarize recent findings on epigenetic adaptation to high altitude living.  The authors give an overview over epigenetic machanisms under hypoxia of humans and animals. Further, epigenetic implications of high altitude induced pulmonary hypertension are discussed. Finally the authors present a concept which uses Exosomes to characterize the Exposome of high altitude living. To me this well presented idea is of high interest and a very nice approach for further studies.  

In general the paper is well written and easy to follow. The authors focus on central mechanisms without getting lost in to many details. To me the review is suitable for publication. I only have some very minor points to address:

  1. Please check if you want to use a space between number and unit or not (e.g. line 48 8000m and line 54 2500 m)
  2. Check your abbreviations (e.g. line 175 HIF-1a/ line 217 HIF1a) 
  3. Some abbreviations are introduced twice (e.g. HIF and EPAS1). For EPAS1 you can decide if you use EPAS1 or HIF-2a in the manuscript.
  4. In line 454 a DOI can be found. Should this be converted to a reference?
  5. Figure 2: Maybe you can mention the Exposomes in the text. Possibly, it could be confusing only to have them in the figure.

Author Response

  1. Please check if you want to use a space between number and unit or not (e.g. line 48 8000m and line 54 2500 m)

It is now corrected-now there is no space between meter and the number in the entire manuscript!

Check your abbreviations (e.g. line 175 HIF-1a/ line 217 HIF1a)

Corrected! Now it’s uniform throughout the review.

  1. Some abbreviations are introduced twice (e.g. HIF and EPAS1). For EPAS1 you can decide if you use EPAS1 or HIF-2a in the manuscript.

I have corrected this throughout the review for many different abbreviations, thank you! I have used EPAS1 everywhere instead of HIF-2a.

  1. In line 454 a DOI can be found. Should this be converted to a reference?

Yes, I have added the doi as reference (Ref:101, Line:471)

Figure 2: Maybe you can mention the Exposomes in the text. Possibly, it could be confusing only to have them in the figure.

I have now defined exposomes in the very beginning of the review (Please see: Line 34-37)

Reviewer 2 Report

Only a few minor comments.

1. On page 6, at line 305,  "---LINE-1. Long Interspersed Nuclear Element-1 (LINE-1)",  should be corrected. Don't use abbreviations first.

2. Similarly,  on page 6 at line 309. "---EPAS1 methylation. Endothelial PAS domain protein 1 (EPAS1)", should be corrected.

3. On page 10 at line 454, "(DOI 454 https://doi.org/10.1007/978-3-642-37078-6_206)" should be numbered.

4. It is well known, that Sirt1 and p53 are closely related. Please discuss and add more references regarding these pathways (On page 8 at line 379-387).

Author Response

  1. On page 6, at line 305,  "---LINE-1. Long Interspersed Nuclear Element-1 (LINE-1)",  should be corrected. Don't use abbreviations first.

It stands corrected now!

  1. Similarly,  on page 6 at line 309. "---EPAS1 methylation. Endothelial PAS domain protein 1 (EPAS1)", should be corrected.

I have corrected this, too! 

  1. On page 10 at line 454, "(DOI 454 https://doi.org/10.1007/978-3-642-37078-6_206)" should be numbered.

I have numbered it (Ref:101, Line:471) 

  1. It is well known, that Sirt1 and p53 are closely related. Please discuss and add more references regarding these pathways (On page 8 at line 379-387).

Thank you for the suggestion ! Please see Lines: 220-227) I have included this under “Epigenetics of hypoxia” section.

Reviewer 3 Report

The paper represents an interesting review about epigenetic mechanisms involved in hypoxia and their role in adaptation to high altitude in humans.

Author have been able to highligth very well the role of epigenetics in this particular situation. Moreover they proposed exosomes as tools to study epigenetic mechanisms and they used also the term exposome.

Authors used the term exposome but they should also provide a description and clarify what they intend to describe with exposome ( only hypoxia? solar radiation, diet etc)

This idea to use exosomes to study epigenetics is interesting and I agree with authors that is a promising topic although I think that considering the paper, the title might be misleading because there are not practical examples or results in this sense. Authors provide an interesting idea but there are not results published as examples.

Authors should also describe results about epigenetic clock and hypoxia.

Authors divided the text in several chapters but in my opinion there is not a real difference between the second point "epigenetic mechanism in hypoxic adaptation " and the the following point Epigenetics of animals at high altitude where there is a similar description of epigenetic changes associated with hypoxia.  

Authors highlighted the advantages to use exosomes but it would be important to describe also potential limitations.

Author Response

The paper represents an interesting review about epigenetic mechanisms involved in hypoxia and their role in adaptation to high altitude in humans.

Author have been able to highligth very well the role of epigenetics in this particular situation. Moreover they proposed exosomes as tools to study epigenetic mechanisms and they used also the term exposome.

Authors used the term exposome but they should also provide a description and clarify what they intend to describe with exposome ( only hypoxia? solar radiation, diet etc)

We have included a definition for exposomes in the very beginning of the review now ((Please see: Line 34-37)

This idea to use exosomes to study epigenetics is interesting and I agree with authors that is a promising topic although I think that considering the paper, the title might be misleading because there are not practical examples or results in this sense. Authors provide an interesting idea but there are not results published as examples.

The very reason that there are no studies on exosomes in the context of epigenetic adaptive mechanisms in High- Altitude Humans prompted us to write this review.

Authors should also describe results about epigenetic clock and hypoxia.

Thanks so much for the suggestion! I have included few lines regarding epigenetic clock in the context of hypoxia in Lines 319-327.

Authors divided the text in several chapters but in my opinion there is not a real difference between the second point "epigenetic mechanism in hypoxic adaptation " and the the following point Epigenetics of animals at high altitude where there is a similar description of epigenetic changes associated with hypoxia.  

Epigenetics of hypoxia – This section deals mainly with HIF and how epigenetics modifies HIF activity in response to hypoxia. This part mostly contains information derived from in vitro models.

Epigenetic adaptation of animals to high-altitude – This section mainly deals with the epigenetic changes that happen in animals, including humans, during real life high-altitude exposure rather than hypoxia alone.

I tried to give a better title now. And included all animal data (in vivo data) in the “Epigenetic adaptation of animals to high-altitude” subheading.

Authors highlighted the advantages to use exosomes but it would be important to describe also potential limitations.

Thanks for this suggestion! Please see Lines: 651-658)